# Personality-Traits Taxonomy and Operational and Environmental Performance: A Cross-Sectional Analysis of Small and Medium Scale Manufacturing Enterprises

Chinyerem Grace Adeniji, Odunayo Paul Salau, Opeyemi Olunike Joel, Oluwakemi Oluwafunmilayo Onayemi * and Oluwakemi Rebecca Alake

Business, Entrepreneurship and Innovation Cluster, Department of Business Management, Covenant University, Ota 12104, Ogun State, Nigeria; chinyere.adeniji@covenantuniversity.edu.ng (C.G.A.); odunayo.salau@covenantuniversity.edu.ng (O.P.S.); opeyemi.joel@covenantuniversity.edu.ng (O.O.J.); oluwakemi.alake@covenantuniversity.edu.ng (O.R.A.)
* Correspondence: oluwakemi.onayemi@covenantuniversity.edu.ng

**Abstract:** This research aimed to assess the operational and environmental performance of small- and medium-sized enterprises (SMEs) in Nigeria in relation to their adoption of personality-traits taxonomy (i.e., conscientiousness, openness to experience, extraversion, neuroticism or emotional resilience and agreeableness). The survey-based study involved the entire population of SME operators in South West Nigeria, totaling 1842 individuals (i.e., entrepreneurs). Through the use of stratified random-selection methods, a sample size of 420 was chosen. Data was collected, and both structural and measurement models were used to assess it. The results of the study demonstrate that personality traits have a significant influence on how successful SMEs function. The results also demonstrate that personality traits have a positive impact on SMEs' environmental performance. These findings suggest that sustaining operational and environmental performance may be accomplished by resource allocation, promoting diversity and inclusion, and employing trustworthy assessment methods. To enhance both their overall operational performance and their environmental performance in particular, the study advises SME operators in Nigeria to undertake proper management of personality traits for sustainability. The paper recommends for greater research on this subject and emphasises the necessity of understanding how personality factors impact operational and environmental performance in Nigerian SMEs.

**Keywords:** agreeableness; conscientiousness; environmental performance; extraversion; neuroticism; openness to experience; operational performance

## 1. Introduction

Managing personality traits for sustainability in SMEs presents significant challenges due to resource limitations and the challenge of aligning unique personality traits with sustainability goals. Small- and medium-sized enterprises (SMEs) usually have resource constraints, such as budgetary constraints and a lack of HR personnel, which makes it challenging to carry out a comprehensive talent-management processes focused on detecting and strengthening employees' personality traits. This limitation could make it more challenging for SMEs to identify and nurture staff members who demonstrate traits compatible with sustainable operations [1].

Additionally, aligning individual personality traits with sustainability goals can be challenging. Certain traits, such as conscientiousness, pro-environmental values, and openness to change, may be more conducive to adopting and promoting sustainable practices [2,3]. However, it is crucial to recognise that individuals possess a range of personality traits, and not all traits may inherently support sustainability efforts. Balancing

the diverse set of traits within an organisation and ensuring that individuals with relevant traits are in positions that influence sustainability decision making can be complex.

The subjective nature of assessing personality traits adds another layer of complexity. Evaluating personality traits often relies on self-report questionnaires or supervisor assessments, which can be influenced by biases, inaccuracies, and individual perceptions [4]. These challenges may result in incomplete or unreliable assessments of personality traits, limiting the organisation's ability to effectively manage and leverage these traits for operational and environmental performance.

Numerous significant factors emerge when personality traits are evaluated for their influence on the operational and environmental performance of SMEs in Nigeria. Firstly, given that cultural norms and values can influence how characteristics are expressed and understood, measuring and managing personality traits across distinct SMEs in Nigeria may be more challenging due to the country's cultural diversity and regional disparities [5]. Second, it could be challenging to find and develop individuals who have personality traits that are advantageous to operational and environmental performance due to the Nigerian SME sector's limited access to resources and training opportunities [6]. Furthermore, it can be challenging to effectively identify people who have the traits required for long-term success due to the subjective nature of personality-characteristic evaluation, which can lead to biases and errors [7]. Additionally, the lack of awareness and understanding of the role of personality traits in driving operational and environmental performance within SMEs may hinder their adoption and integration into management practices [6].

The limited research specifically focusing on personality traits and their impact on operational and environmental performance in Nigerian SMEs further complicates the identification and understanding of these challenges. Lastly, the complex and evolving regulatory landscape in Nigeria, including environmental policies and compliance requirements, may pose additional challenges in leveraging personality traits to drive sustainable practices within SMEs [8]. Addressing these challenges requires targeted research, cultural sensitivity, resource provision, training initiatives, and the integration of personality-trait considerations into sustainable management practices within Nigerian SMEs.

Managing personality traits for sustainability in SMEs poses several challenges. Firstly, SMEs often have limited resources and capacity for comprehensive talent-management practices, including assessing and developing employees' personality traits [9]. Secondly, there can be difficulties in aligning individual personality traits with sustainability goals, as certain traits may be more conducive to sustainability practices than others [10]. The subjective nature of assessing personality traits and the potential for biases in the evaluation process add further complexity to effectively managing and leveraging these traits for sustainable outcomes [4,7]. A lack of acknowledgment and a lack of understanding of the significance of environmental sustainability via such practices have hindered SMEs' efforts towards managing personality traits, despite the fact that they are the driving force behind the expansion of the manufacturing sector in Nigeria. Therefore, the purpose of this study is to investigate the causes of this gap and to suggest ways to encourage SMEs in Nigeria to embrace personality-traits taxonomy. In this study, a cross-sectional analysis was conducted to investigate the relationship between personality traits and operational and environmental performance in small- and medium-scale manufacturing enterprises. The study design incorporated surveys to gather personality-trait data from employees and environmental performance data from the companies, which were subsequently analysed using regression analysis to assess potential associations between personality traits and environmental performance outcomes. Hence, this study was conducted to fill the knowledge gap by addressing the following research questions:

i.   To what extent does personality-traits taxonomy affect SMEs' environmental performance?
ii.  In what way does personality-traits taxonomy affect the operational performance of SME's?

## 2. Literature Review

### 2.1. Conceptual Clarification

2.1.1. Personality Trait

Personality traits are enduring patterns of emotions, ideas, and behaviors that are mostly constant across time and across circumstances. They are distinguishing characteristics that influence how people perceive and interact with their surroundings [11]. The various facets that personality traits might encompass include extraversion, conscientiousness, openness to new experiences, agreeableness, and emotional stability, to name just a few. The aforementioned elements are part of the Five-Factor Model (FFM), sometimes known as the Big Five, which is a popular framework for examining personality traits. The FFM provides a comprehensive framework for analyzing and categorizing different personality traits. For instance, Barrick and Mount's meta-analysis from 1991 confirmed the value of the Big Five characteristics in predicting work performance. In a similar vein, ref. Stern, Dietz, Abel, Guagnano and Kalof, (1999) [12] discussed how personality traits, in particular being open to new experiences, may influence creativity and innovation. Personality traits are crucial components of someone's psychological makeup and may have an impact on a range of aspects of their life, including their behaviour, attitudes, and performance in a number of settings, including their job.

i.  Extraversion: Extraversion, one of the core personality traits, is crucial in determining how people interact with the outside world. Extraverts are sometimes characterised as outgoing, sociable, and energetic people who prefer to be around other people [13]. Extraversion is a personality attribute that presents people with a unique mix of advantages and difficulties. While extraverts are frequently praised for their outgoing personalities, boundless energy, and capacity for human connection, it is crucial to understand that introversion also possesses key advantages. Ernawati [14] postulated that extraverted people are more likely to be gregarious and outgoing, which is advantageous for SMEs in terms of networking and connection development. They could discover it is simpler to establish relationships with possible partners, consumers, clients, and investors, which might result in business expansion and opportunities.

ii.  Conscientiousness: This is to be well-organised, accountable, trustworthy, and goal-oriented. Conscientiousness may have a major and generally beneficial impact on how well small- and medium-sized enterprises (SMEs) function. Conscientiousness may improve dependability, organisation, attention to detail, and general efficiency, which can have a positive and significant influence on SME success [15]. It helps create a trustworthy and accountable workplace culture, which may enhance client happiness, financial security, and general success. Conscientiousness may have a variety of beneficial influences on SME performance, but it should ideally be balanced with other characteristics within the organisation. Wilmot and Ones posited that an excessive emphasis on conscientiousness alone may result in rigidity or reluctance to change, which can be harmful in businesses that are dynamic or forward-thinking. Therefore, in SMEs, a team with a diverse set of personality types might typically be the most productive.

iii.  Openness to new experiences: It displays a person's propensity to be inquisitive, creative, open-minded, and responsive to new concepts and experiences. People with more openness tend to be creative, adventurous, and eager to explore new areas. SMEs frequently deal with quickly shifting market conditions. Due to their adaptability and familiarity with change, open people are better able to deal with uncertainty and shift course when necessary [16]. They can aid SMEs in capturing new possibilities and successfully addressing obstacles. Kerr [17] alluded that a personality trait that has a substantial impact on the performance of small- and medium-sized enterprises is openness to new experiences. It encourages creativity, adaptability, and a customer-centric mindset, all of which SMEs need to succeed in the fast-paced business world of today. To reach its greatest potential, it should be combined with other qualities and be well controlled. SMEs are ultimately more likely to succeed, develop, and boost

the local economy when they embrace and foster an openness to new experiences. Agyei [18] showed that although being open to new experiences may enhance SMEs' performance in many ways, it is important to understand that this quality may not always be advantageous. An excessive emphasis on openness without taking other qualities like conscientiousness or risk management into account might result in rash decisions or a lack of concentration. Success in an organisation typically depends on striking a balance between various personality types.

iv.    Agreeableness: Agreeableness is a personality trait that has a big impact on how well SMEs function. As one of the five main personality qualities in the Five Factor Model of Personality, agreeableness represents a person's propensity for cooperation, friendship, and compassion [19]. Characteristics like empathy, friendliness, and a desire to work together define agreeableness. People that score highly on the agreeableness scale are frequently characterised as kind, thoughtful, and warm [20]. This personality trait can have a significant effect on several facets of organisational success in the context of SMEs. Employees with high agreeableness can develop good working relationships, boost communication, and promote a happy workplace culture in SMEs where cooperation is frequently essential. This culture of cooperation may boost creativity and productivity. Anwar [21] acknowledged that although agreeableness has several benefits for SMEs, it is important to understand that this personality attribute may not always be the most appropriate in all circumstances. An overemphasis on agreeableness may result in difficulties such as the inability to bargain assertively when necessary, or a predisposition to avoid confrontation at all costs. For an organisation to succeed holistically, a blend of many personality qualities, such as assertiveness and conscientiousness, is commonly required.

v.    Neuroticism: A crucial component of emotional intelligence is emotional stability, which includes the capacity to successfully control and regulate one's emotions. People who are emotionally stable typically maintain their composure under stress, demonstrate self-control, and bounce back swiftly after setbacks or pressures. This personality characteristic can significantly affect organisational effectiveness in the setting of SMEs. People who are emotionally stable are better able to make thoughtful judgements, especially in trying circumstances [21]. Leaders and staff who display emotional stability can evaluate events objectively and select the most appropriate courses of action in SMEs, where resource restrictions and uncertainty are widespread. SMEs frequently deal with short deadlines, financial pressures, and market fluctuations. Employees that are emotionally secure are less prone to experience stress and burnout. They are able to stay focused, handle pressure from their jobs well, and maintain high levels of productivity over time. A crucial personality quality that profoundly affects the functioning of small- and medium-sized enterprises is emotional stability [22]. Employees and executives who are emotionally stable contribute to logical decision making, effective conflict resolution, adaptation to change, and stress-resilience skills that are essential in the fast-paced environment of SMEs. Emotional stability may be a significant asset that fosters organisational success, longevity, and growth when it is effectively handled and paired with other personality attributes. SMEs that place a high priority on emotional intelligence are better equipped to deal with obstacles and take advantage of opportunities, eventually promoting economic development.

### 2.1.2. Operational Performance

Operational performance is the effectiveness and efficiency of an organisation's internal processes and operations in achieving its strategic goals [23]. It comprises a variety of metrics, such as those that deal with quality, cost management, productivity, and customer satisfaction [24]. The ability to manufacture goods or render services quickly, superbly, and affordably is known as exceptional operational performance. Ologhodo [25] posited that any successful organisation is built on its operational performance. It includes the

efficacy and efficiency with which a business controls its resources, procedures, and operations to accomplish its strategic goals. The capacity to continuously produce excellent operational performance is essential for not just surviving but also thriving in the highly competitive corporate climate of today. Operational performance refers to the evaluation of a company's capacity for cost-effectively producing goods or providing services, while maximising resources and satisfying consumer demands [26].

2.1.3. Environmental Performance

Environmental performance, on the other hand, focuses on the impact an organisation's operations have on the environment and their sustainability. It comprises evaluating and improving approaches to waste management, pollution prevention, and adherence to environmental standards and legislation [27]. Strong environmental performers make an attempt to limit negative environmental consequences, reduce their ecological footprint, and save resources [28]. An important component of contemporary business practises is environmental performance, sometimes known as environmental sustainability. It relates to an organisation's attempts to lessen its ecological footprint, lessen detrimental environmental effects, and help preserve the environment. Adebayo [29] opined that focusing on environmental performance has evolved from being a moral need to being a tactical necessity in a world that is becoming increasingly worried about climate change and environmental deterioration. Environmental performance is no longer an afterthought for businesses; rather, it now serves as a cornerstone of ethical and sustainable corporate behaviour. Organisations can not only lessen their environmental impact by putting a high priority on resource efficiency, emissions reduction, waste management, and biodiversity protection, but they can also reap a lot of advantages, including improved reputation, cost savings, and long-term competitiveness. Chuang [30] attested that in an increasingly environmentally concerned society, adopting environmental sustainability is not only morally required but also a smart tactical move for enterprises. By doing this, businesses may ensure their own long-term profitability while helping to make the earth a better place.

2.1.4. Personality Trait, Operational and Environmental Performance

Research on the effect of personality traits on the operational and environmental performance of SMEs is limited, making it challenging to draw definitive conclusions. While some studies have explored the relationship between personality traits and organisational outcomes, such as job performance and innovation [12,28], their direct impact on operational and environmental performance in SMEs remains unclear. Additionally, the existing research often focuses on larger organisations or broader contexts, making it difficult to generalise findings to the specific context of SMEs in terms of their operations and environmental practices. Furthermore, studies examining the role of personality traits in sustainability-related behaviours have shown that traits like conscientiousness and pro-environmental values can influence individuals' engagement in environmentally friendly behaviours [3,12]. However, the direct link between these traits and operational and environmental performance in SMEs requires further investigation. There is a need for more targeted research that specifically examines the relationship between personality traits and the operational and environmental performance of SMEs. Longitudinal studies that track performance outcomes over time and mixed-methods approaches combining quantitative measures with qualitative insights would provide a more comprehensive understanding. Additionally, comparative studies across different industries and organisational sizes can shed light on potential variations and contextual factors that may influence this relationship.

*2.2. Hypotheses Development*

2.2.1. Personality Trait and Operational Performance of SMEs

The impact of personality traits on SME operational performance has received limited attention in the existing literature. However, studies in broader organisational contexts

have suggested that certain traits, such as conscientiousness and openness to experience, can influence job performance and innovation [8,11]. These traits may indirectly contribute to SME operational performance by affecting employee behaviour, decision-making, and adaptability to change. Nonetheless, further research is needed to specifically examine the relationship between personality traits and SME operational performance, considering the unique challenges and dynamics within this context. Based on the proposition above, the following hypothesis was formulated:

**Hypothesis 1 (H₁).** *Personality traits taxonomy has a significant relationship with the operational performance of SMEs'.*

### 2.2.2. Personality Trait and SME Environmental Performance

Research on the effect of personality traits on SME environmental performance suggests that certain traits, such as conscientiousness and pro-environmental values, can influence individuals' engagement in environmentally friendly behaviours, and the involvement of small- and medium-sized enterprises (SMEs) for the achievement of sustainable development is of paramount importance [3,12,31]. These traits may indirectly contribute to SME environmental performance by shaping employees' attitudes, behaviours, and decision-making processes related to sustainability practices. However, further research is needed to establish a more definitive and comprehensive understanding of the specific personality traits that have the most significant impact on SME environmental performance. Based on the proposition above, the following hypothesis was formulated:

**Hypothesis 2 (H₂).** *Personality-traits taxonomy has a significant effect on SMEs' environmental performance.*

### 2.3. Conceptual Framework

Figure 1 shows the hypothetical model of the association between personality traits, operational and environmental performance of SMEs.

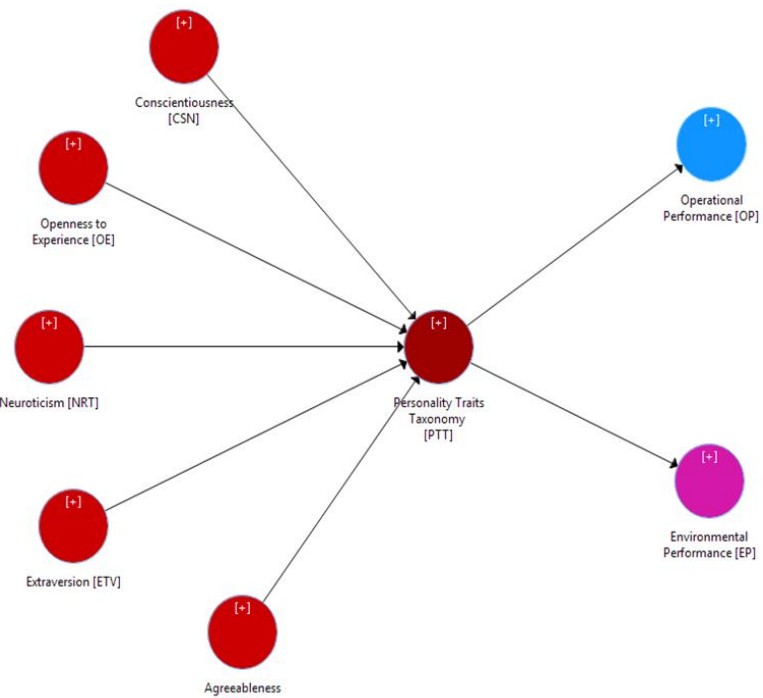

**Figure 1.** Hypothetical model of the association between personality trait, operational and environmental performance of SMEs.

Figure 1 presents a hypothetical model that illustrates the envisioned relationship between personality traits, operational and environmental performance of small- and medium-sized enterprises (SMEs). In this model, personality traits are positioned as a central factor influencing the operational and environmental performance of these organisations. The arrows connecting the two constructs represent the proposed direction of influence, suggesting that specific personality traits may exert a direct or indirect impact on the operational and environmental performance of SMEs. The figure serves as a visual representation of the theoretical framework guiding the research, which aims to explore and validate these connections, providing valuable insights into the interplay between individual personality attributes and operational-environmental performance efforts of SMEs.

### 2.4. Theoretical Justifications

This study is hinged on The Big Five model and Contingency Theory. The Contingency Theory, propounded by Donaldson in 1969, assumes that there is no universal or one-size-fits-all approach to management, and instead posits that the most effective management style is contingent upon various factors within a specific organisational context. This theory emphasises that the optimal management style can vary based on factors such as the organisation's size, structure, technology, and external environment. The theory acknowledges that different situations and conditions call for different approaches to leadership, thereby rejecting the notion of a single best way to manage organisations. In the context of the study on personality-traits taxonomy and operational and environmental performance, this theory suggests that the influence of personality traits on environmental performance may vary depending on the unique characteristics and circumstances of each small- and medium-scale manufacturing enterprise, aligning with the contingency perspective that underscores the importance of adapting management strategies to fit the specific context.

The Big Five model offers a framework for comprehending how particular personality traits relate to the effectiveness of the operational environment [32]. Openness to new experiences, conscientiousness, extraversion, agreeableness, and neuroticism are just a few of the personality traits that can affect an organisation's environmental practices and results. On the other hand, the relationship between personality traits and performance in an operational environment is influenced by contextual factors, according to contingency theory [33]. These elements might consist of the organisation's size, structure, culture, sector, and external pressures. Depending on the circumstances, different combinations of personality traits may be more effective. In a highly regulated industry, for example, companies may benefit from diligent and detail-oriented employees, whereas individuals who are open to experience and creative thinking may be crucial for advancing sustainability efforts in a field that is changing quickly. By combining the lessons from the Big Five model and Contingency Theory, researchers and practitioners can gain a better understanding of how personality traits interact with organisational contexts to affect operational and environmental performance [34]. This knowledge can be used to create specialised strategies and interventions to enhance sustainability practices in small- and medium-sized manufacturing enterprises.

The Contingency Theory's implications for the study on personality-traits taxonomy and operational and environmental performance of SMEs are significant. As this theory highlights the importance of adapting management approaches to the unique characteristics of each organisation, it underscores that the influence of personality traits on environmental performance in SMEs may not follow a uniform pattern. Different SMEs may possess varying organisational structures, cultures, and external contexts that require distinct leadership styles. Therefore, when analysing the relationship between personality traits and environmental performance in small- and medium-scale manufacturing enterprises, it is crucial to consider the specific contingencies and contextual factors that might moderate this relationship. The Contingency Theory prompts researchers and practitioners to recognize

that a one-size-fits-all approach may not be suitable for understanding or improving environmental performance within SMEs, emphasizing the need for tailored strategies that align with the particular circumstances of each organisation.

## 3. Materials and Methods

This study, which concentrated on the registered SME manufacturing operators in South West, Nigeria, used a survey research design (i.e., descriptive design). A sample size of 420 SMEs was chosen using a stratified-random-selection approach from the study's population of 1842 SMEs. Using the PAC (SPSS version 26) programme, the sample was dispersed throughout South West Nigeria, and the data analysis was done using both descriptive and inferential statistical approaches. Specifically, structural and measurement models (SEM_SmartPLS, version 3) were used to analyse the collected data. A standardised 5-point Likert scale questionnaire was used to collect the data. The validation of the questionnaire was ensured using a representative sample, demonstrating adequate reliability and validity. This is essential for determining how strongly the participants agree with the items on the research instrument [35].

The study utilised five proxies of personality-traits taxonomy (PTT) (i.e., conscientiousness, openness to experience, extraversion, neuroticism or emotional resilience, and agreeableness). The ratings for all observed variables were assessed on a scale ranging from 1 to 5. The questionnaire used a five-scale Likert format to capture the precise level of attention and responses to the research items. 1 = Strongly Agree (SA), 2 = Agree (A), 3 = Undecided (U), 4 = Disagree (D), and 5 = Strongly Disagree (SD). The Likert scale pattern of the research instrument guides the respondents' selection process. Hence, the questions compiled were grouped as follows:

i.   Section A: Items on respondents' demographic profile (Items 1–4).
ii.  Section B: Items on personality-traits taxonomy and operational and environmental performance (Items 1–25). This is presented in Table 1.

**Table 1.** Items in the questionnaire and their sources.

| S/N | Constructs | Variables | Number of Items | Sources |
|---|---|---|---|---|
| 1 | Personality Traits Taxonomy | Conscientiousness (Items 1–3) | 3 | Sarwoko [22], Anwar [21], Judge [7], Sleep [32], Feher [34], Sarwoko [22] |
| | | Openness to experience (Items 4–6) | 3 | |
| | | Neuroticism (Items 7–9) | 3 | |
| | | Extraversion (Items 10–12) | 3 | |
| | | Agreeableness (Items 13–15) | 3 | |
| 2 | Operational | Environmental Performance | 10 | Robertson [4], Karpinski [3], Nwaogbe [26], Chuang [30], Rehman [28], Adebayo [29], Awolusi [24], Mbah [23] |

Each multivariate premise was verified twice to ensure accuracy before conducting the analysis. The analysis involved the use of *p*-values, Rsquare values, path coefficients, and t-statistics to examine the data. Figure 2 presents the path coefficient, which determines the strength and direction of the relationship between personality-trait practices and operational and environmental performance. The *p*-value, also shown in Figure 3, indicates the level of significance. It also displays the path's coefficient analysis, illustrating the interaction among the variables in the model.

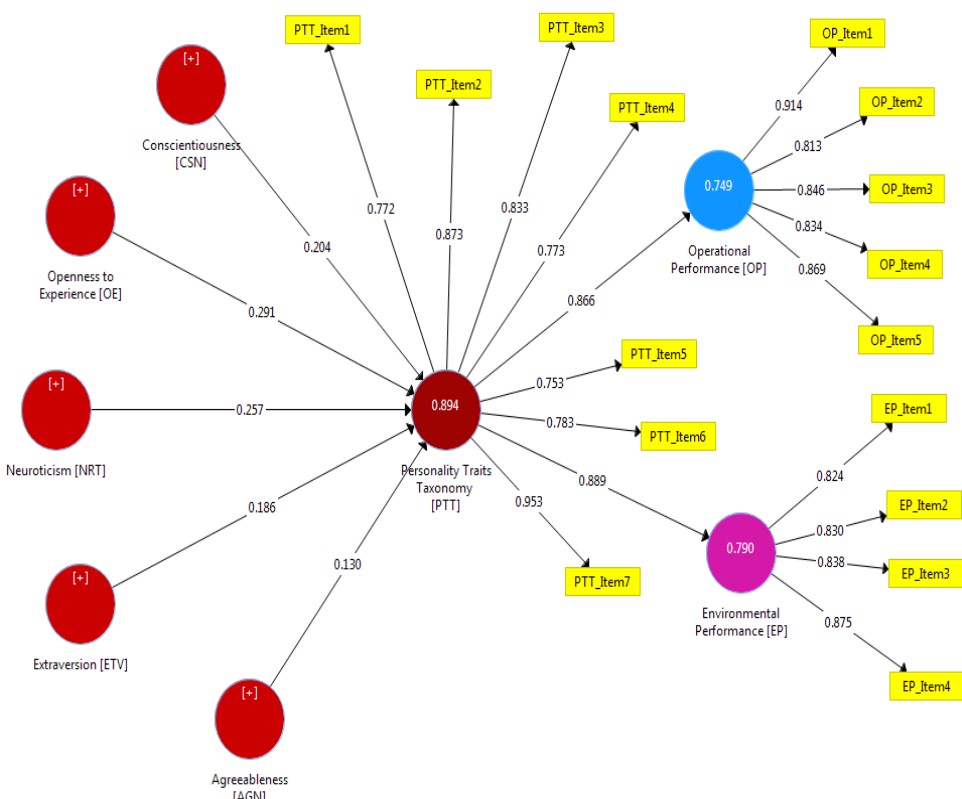

**Figure 2.** SEM path's coefficient.

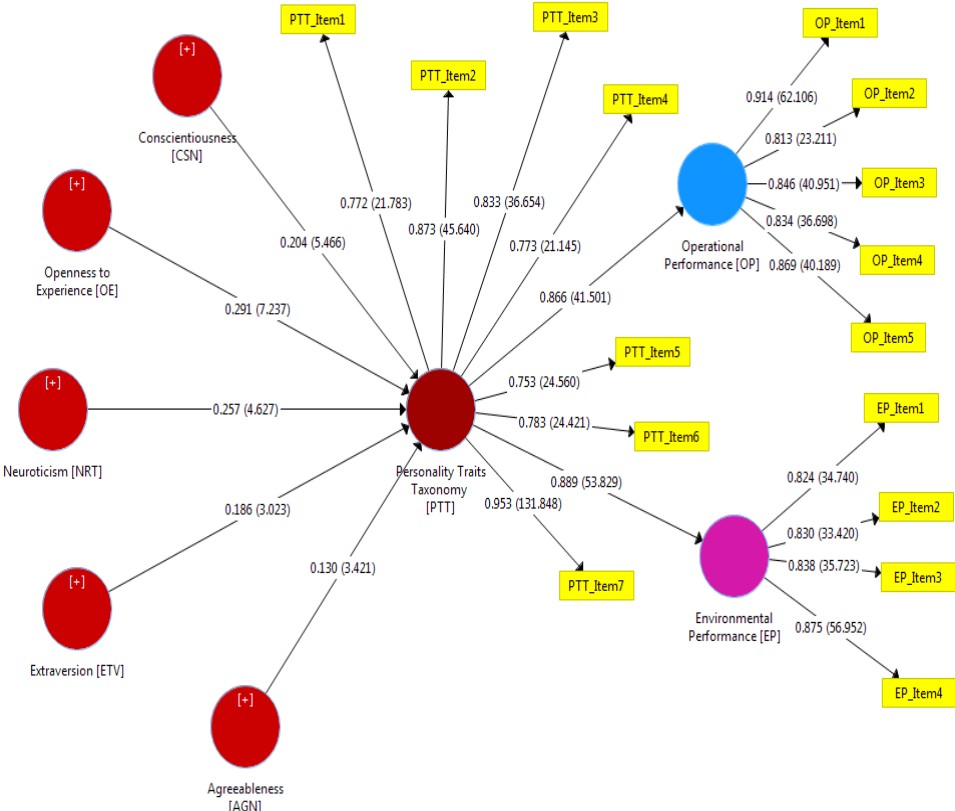

**Figure 3.** SEM path's coefficient and Tvalues.

### 3.1. Response Rate

The research distributed four hundred and twenty (420) copies of the questionnaire to respondents that participated in the study. The study used three hundred and eighty-eight (388) copies of returned questionnaires to conduct the analysis. Table 2 illustrates that this equates a response rate of 92.4%.

**Table 2.** Administered questionnaire response rate.

| Response Rate | Number | Percentage |
| --- | --- | --- |
| Total number of questionnaires distributed | 420 | 100.0% |
| Copies returned | 388 | 92.4% |
| Unreturned copies | 32 | 7.6% |

Table 2 provides information on the distribution and return rates of questionnaires in a survey. It shows that 420 questionnaires were distributed in total, and 388 of them were returned, representing a return rate of 92.4%. This high return rate indicates a strong level of engagement and cooperation from the respondents, which enhances the reliability and representativeness of the data collected. The 7.6% of unreturned copies may suggest that some respondents were not interested or unable to participate, but the high response rate still provides a robust dataset for analysis, allowing for meaningful insights and implications to be drawn from the survey results.

### 3.2. Demographic Profiles of Respondents

Table 3 showed the demographic characteristics of the study participants. This section provides a platform for understanding the demographics of participants (SMEs' business owners and managers) in the survey. The data analysis sheds light on the gender distribution, age demographics, educational backgrounds, and characteristics of SMEs' business owners and managers.

**Gender**: According to the data, women make up 54.9% of all business managers and owners in SMEs, which is slightly higher than the national average. On the other hand, male business owners and managers make up 45.1% of the sample. This suggests that the gender distribution in SME leadership is fairly even. According to the data, there is a slight gender disparity in the number of male and female SME business managers and owners. This exemplifies how women are increasingly taking on leadership and entrepreneurial roles within SMEs. As a result, SMEs ought to support gender diversity and an environment that encourages equal opportunities for men and women.

**Age**: The age distribution of SME business managers and owners provides some fascinating insights. The majority of the sample, or 46.6%, is made up of people between the ages of 31 and 40. It seems that middle-aged people are actively involved in owning and managing SMEs as a result. The next most significant age group is between the ages of 17 and 30, which makes up 24.2% of the total. This highlights the fact that there are young managers and business owners in the SME sector. The sample also includes 17.3% of people between the ages of 41 and 50 and 11.9% of people over the age of 51. The majority of SMEs' business owners and managers are between the ages of 31 and 40, indicating that people in the prime of their careers are actively involved in SMEs. This suggests that leaders of SMEs are frequently seasoned individuals who have a good balance of enthusiasm and knowledge. However, the fact that there are younger managers and business owners in the 17–30 age range highlights how important it is to foster and support young people's entrepreneurial spirit. It implies that mentorship programmes, resources, and an enabling ecosystem are required to support young entrepreneurs.

**Academic qualifications**: We can see a wide range of qualifications when we look at the educational background of SME business owners and managers. The highest percentage, 34.0%, report having an SSCE as their highest academic achievement, showing that a sizable portion of SME owners and managers have finished their secondary education.

Additionally, 29.1% of people hold an NCE/OND, which is equivalent to a diploma or level of training in vocational studies. A sizeable percentage (27.1%) of the population has an HND/B.Sc./B.Ed., demonstrating a higher level of education attained through technical colleges or universities. Last but not least, 9.8% of people have an M.SC./MBA, showing a higher percentage of people with postgraduate degrees in management or specialised fields. The distribution of educational backgrounds among SME business managers and owners demonstrates a wide range of educational backgrounds. It is important to note that a sizeable percentage of them also hold higher credentials like an HND/B.Sc./B.Ed. and an M.SC./MBA, even though a sizable percentage of them hold an SSCE or NCE/OND, indicating a range of credentials from secondary education to vocational or diploma programmes. According to this, effective SME leaders come from a range of academic backgrounds. It suggests that both the theoretical knowledge obtained through higher education and the practical skills obtained through experience have an impact on how well SMEs perform. This highlights the importance of multidimensionally focusing on entrepreneurship education and skill development.

**Table 3.** Demographic profiles (n = 388).

| Variables | Frequency | Percentage |
|---|---|---|
| Gender | | |
| Male | 175 | 45.1 |
| Female | 213 | 54.9 |
| Total | 388 | 100% |
| Age | | |
| 17–30 | 94 | 24.2 |
| 31–40 | 181 | 46.6 |
| 41–50 | 67 | 17.3 |
| 51 and above | 46 | 11.9 |
| Total | 388 | 100% |
| Academic Qualifications | | |
| SSCE | 132 | 34.0 |
| NCE/OND | 113 | 29.1 |
| HND/B.Sc./B.Edu. | 105 | 27.1 |
| M.SC/MBA | 38 | 9.8 |
| Total | 388 | 100% |
| Nature of SMEs | | |
| Manufacturing | 56 | 14.4 |
| Mining and Quarry | 47 | 12.1 |
| Manufacturing | 86 | 22.2 |
| Wholesale Trade | 80 | 20.6 |
| Retail Trade | 93 | 24.0 |
| Others | 26 | 6.7 |
| Total | 388 | 100% |

**Nature of SMEs**: The data demonstrates the distribution of SMEs across different industries. The largest category is retail trade, accounting for 24.0% of the sample. This demonstrates that a significant portion of SMEs engage in retail activities. Manufacturing represents another significant industry and accounts for 22.2% of the sample. At 20.6%, wholesale trade comes in second. The presence of SMEs (12.1%) in the mining and quarrying sectors should not be overlooked. A total of 6.7% of the sample is made up of SMEs in sectors that are not mentioned in the data under the "Others" category. The distribution of

SMEs across various industries provides insight into the diversification of entrepreneurship. Given their popularity, the manufacturing, retail, wholesale, and mining and quarry sectors are likely to see growth and opportunity. SMEs working in these industries should put innovation, quality, and competitiveness as their top priorities to succeed in the market. The fact that SMEs exist in other less-described sectors is another sign of the need for more research and understanding to better support and address the particular challenges faced by these businesses.

### 3.3. Descriptive Statistics

In this section, descriptive statistics are presented to illustrate the frequency distribution of the specific items used to measure personality-traits taxonomy and operational and environmental performance. Frequency distribution is used to code and interpret the data acquired and processed. All measurements were available in five Likert scales, i.e., strongly agreed–strongly disagree. Table 4 provides further information on personality-traits taxonomy.

**Table 4.** Personality-traits taxonomy (PTT) [n = 388].

| SN | ITEMS | SA | A | U | D | SD | Mean | SD |
|---|---|---|---|---|---|---|---|---|
| **Openness to Experience:** | | | | | | | | |
| OP1 | I enjoy trying new things and exploring different ideas. | 44% | 30% | 6% | 9% | 11% | 4.346 | 0.7274 |
| OE2 | I am curious and enjoy learning about new subjects. | 34% | 42% | 7% | 10% | 7% | 3.937 | 0.8275 |
| OE3 | I appreciate creativity and artistic expression. | 33% | 41% | 5% | 10% | 11% | 4.082 | 0.7038 |
| **Conscientiousness:** | | | | | | | | |
| CST1 | I am organised and like to plan ahead. | 38% | 40% | 6% | 8% | 8% | 4.104 | 0.7174 |
| CST2 | I pay attention to detail and strive for accuracy in my work. | 32% | 43% | 6% | 10% | 9% | 3.836 | 0.8087 |
| CST3 | I am reliable and always complete tasks on time. | 33% | 40% | 5% | 12% | 10% | 3.904 | 0.6288 |
| **Extraversion:** | | | | | | | | |
| EXT1 | I enjoy being around people and socializing at parties or events. | 35% | 42% | 7% | 9% | 7% | 4.009 | 0.8374 |
| EXT2 | I feel energised by social interactions and being in the company of others. | 40% | 34% | 6% | 11% | 9% | 3.973 | 0.8233 |
| EXT3 | I am comfortable speaking up and expressing my opinions in group settings. | 41% | 30% | 5% | 12% | 12% | 4.103 | 0.9026 |
| **Agreeableness:** | | | | | | | | |
| AGN1 | I am empathetic and considerate towards others' feelings. | 31% | 42% | 8% | 10% | 9% | 4.295 | 0.8286 |
| AGN2 | I enjoy helping others and find it fulfilling to be of service. | 36% | 39% | 8% | 8% | 9% | 3.999 | 0.8196 |
| AGN3 | I prefer cooperation and harmony in my relationships. | 40% | 32% | 8% | 9% | 11% | 4.008 | 0.8826 |
| **Neuroticism:** | | | | | | | | |
| NTC1 | I often worry about things and feel anxious. | 33% | 40% | 7% | 10% | 10% | 4.000 | 0.7497 |
| NTC2 | I am prone to mood swings and emotional ups and downs. | 35% | 40% | 6% | 9% | 10% | 4.207 | 0.8001 |
| NTC3 | I tend to overthink and dwell on the fear of unknown. | 30% | 46% | 6% | 10% | 8% | 4.1467 | 0.8449 |

Table 4 shows the specific items of measurement of **personality-traits taxonomy**.

**Openness to experience**: The data reveal that a sizable portion of respondents (between 33% and 44%) express an openness to novel experiences. This quality is correlated with traits like curiosity, willingness to try new things, and an appreciation for creativity. These results imply the existence of individuals who might be more receptive to adopting

innovative techniques and sustainable projects for SME managers and owners. Utilizing the trait of openness to experience can make it easier to explore new ideas and environmentally friendly technologies or processes.

**Conscientiousness**: The data show that a sizeable percentage of respondents (between 32% and 40%) demonstrate conscientiousness. These people have a reputation for dependability, efficiency, and attention to detail. Employing diligent workers can enhance the performance of the operational environment for SME managers and owners. They can assist organisations in adhering to environmental regulations, implementing sustainable practices, and ensuring that environmental metrics are accurately tracked and reported thanks to their attention to detail and goal-directed behaviour.

**Extraversion**: The data show that between 34% and 42% of respondents (a sizable portion) have extraversion traits. Extraverts find energy in interacting with others and feel free to express their opinions. Managers and business owners of small- and medium-sized enterprises (SME) can create a positive workplace culture by utilising the extraversion trait. These individuals significantly influence the development of the teamwork, cooperation, and effective communication essential for implementing and upholding environmental initiatives.

**Agreeableness**: The data show that between 31% and 40% of respondents have traits associated with agreeableness. Agreeable people value harmony and cooperation in relationships and are sympathetic and helpful. This suggests that managers and owners of SMEs may have access to staff who are more likely to engage in environmentally friendly practices and support sustainability initiatives. By creating a work environment that values cooperation, empathy, and support for one another, a group commitment to operational and environmental performance can be fostered.

**Neuroticism**: According to the data, a sizeable portion of respondents (between 30% and 46%) display neuroticism traits like worrying, mood swings, and overthinking. Although these characteristics might not directly affect the performance of the operational environment, they may nonetheless have an indirect impact on employee happiness and job satisfaction. It is crucial for SME managers and owners to take neuroticism's potential effects on staff participation in environmental initiatives into account. Enhancing employee engagement in sustainability practices can be accomplished by attending to their emotional needs, offering support, and fostering a positive work environment.

Table 5 shows the specific items of measurement and responses of the surveyed individuals regarding various operational environmental practices.

The responses are rated from Strongly Agree (SA) to Strongly Disagree (SD) on a Likert scale. A relatively high implementation rate of energy-saving practices and technologies is indicated by the 41% of respondents who strongly agree that their organisation uses them. However, there is still room for improvement as only 30% of respondents agree, 6% are unsure, and 23% strongly disagree. The survey results reveal areas where the organisation is succeeding and those that need improvement. These data can be used by managers and owners to pinpoint their strengths, build on them, and address their weaknesses in order to improve operational environment performance.

In terms of water usage and effective management, 34% and 42%, respectively, agree that their company has policies in place to cut back on water use. This suggests that water conservation is being approached generally favorably. However, 20% of respondents either strongly disagree or disagree with the statement, which suggests that water-management strategies require more attention. Managers can allocate resources to areas that require improvement based on the survey results. For instance, efforts could be focused on implementing efficient recycling programmes and waste-reduction strategies if waste-management practices had lower agreement rates.

According to 33% of respondents who were asked about waste reduction and recycling, 41% strongly agreed that their organisation actively works to reduce waste generation and puts programmes in place for recycling and waste reduction. This shows a commendable dedication to waste minimization. However, 20% of respondents either disagree

or disagree strongly, suggesting a potential weakness in waste-management procedures. The results suggest that some employees might have different views on environmental practices. Managers can focus on raising awareness, providing training, and fostering an environment of environmental responsibility to ensure that employees are knowledgeable about sustainable practices and actively participate in them.

**Table 5.** Operational and environmental performance (OEP).

| S/N | ITEMS | SA | A | U | D | SD | Mean | SD |
|---|---|---|---|---|---|---|---|---|
| **OEP**1 | Our organisation implements energy-saving practices and technologies in our operations. | 41% | 30% | 6% | 10% | 13% | 4.111 | 0.7937 |
| **OEP2** | We have measures in place to reduce water consumption and promote efficient water management. | 34% | 42% | 4% | 12% | 8% | 3.836 | 0.8284 |
| **OEP3** | Our company actively seeks to minimise waste generation and implements recycling and waste reduction programmes. | 33% | 41% | 7% | 10% | 10% | 4.184 | 0.6927 |
| **OEP4** | We comply with environmental regulations and strive to exceed minimum standards. | 34% | 40% | 5% | 11% | 9% | 4.487 | 0.8498 |
| **OEP5** | We regularly monitor and measure our greenhouse gas emissions to identify areas for reduction. | 31% | 42% | 6% | 11% | 10% | 3.826 | 0.9037 |
| **OEP6** | Our organisation promotes environmentally friendly procurement practices, such as sourcing sustainable materials and suppliers. | 34% | 39% | 7% | 12% | 8% | 4.132 | 0.8673 |

A total of 34% and 40%, respectively, strongly agree that their organisation makes an effort to go above and beyond the minimum requirements for environmental regulations. This exemplifies good rule following. However, 20% of those surveyed said they strongly disagreed with the proposal or thought it needed more work to ensure compliance. Managers can foster a productive work environment that supports sustainable practices by promoting teamwork and getting employees involved in conservation efforts. Setting clear goals, praising and rewarding environmentally conscious behaviour, and encouraging cross-departmental cooperation can all help with this.

According to 31% and 42% of respondents, respectively, their organisation regularly measures emissions to identify areas for reduction. This suggests that proactive emission management is usually practiced. However, 21% of respondents either strongly disagree with the statement or disagree with it, which suggests a monitoring and reduction effort gap. Environmental laws must be adhered to. Managers and owners of SMEs should be aware of relevant regulations and strive to do more than is required. This can include ongoing monitoring, adapting practices to meet new standards, and taking part in industry initiatives and certifications.

A total of 34% of respondents agree, and 39% strongly agree, that their organisation promotes environmentally friendly procurement practices. This implies that the procurement process places a priority on sustainability. A total of 20% of respondents expressed disagreement or strong disagreement, demonstrating that more can be done to choose sustainable suppliers and materials. SMEs can explore green procurement methods and collaborate with sustainable suppliers. Finding environmentally friendly materials, evaluating suppliers' environmental performance, and cultivating partnerships that support the organisation's sustainability goals can all be part of this.

Table 6 provides an assessment of the construct validity and reliability of the variables used in the study. This table offers a comprehensive evaluation of the measurement instruments employed for key constructs, including personality-traits taxonomy (PTT)

and operational and environmental performance (OEP). It highlights the reliability and validity of these constructs, confirming the soundness of the research instruments for further analysis.

**Table 6.** Construct validity and reliability of the variables. Source: Field Survey, 2023.

| Variables | Cronbach Alpha (>0.70) | Composite Reliability (>0.70) | Average Variance (>0.50) |
|---|---|---|---|
| Personality-traits taxonomy (PTT) | 0.785 | 0.848 | 0.614 |
| Operational performance (OP) | 0.807 | 0.873 | 0.633 |
| Environmental performance (EP) | 0.893 | 0.869 | 0.671 |

The table presents the reliability and validity assessment of two key variables in the study: personality-traits taxonomy (PTT) and operational and environmental performance (OEP). Both variables demonstrate high levels of internal consistency, as indicated by their Cronbach Alpha values exceeding the recommended threshold of 0.70. This suggests that the survey items measuring PTT and OEP are reliable and internally consistent. Additionally, the composite reliability scores for both variables also surpass the 0.70 threshold, indicating that the measurement model is robust and reliable. Moreover, the average variance for both PTT and OEP is above 0.50, which signifies good convergent validity, showing that the items within each construct are closely related to each other. These results indicate that the measurement tools for PTT and OEP are both reliable and valid, making them suitable for use in the study, and they provide confidence in the accuracy of the data collected for further analysis.

## 4. Analysis and Discussion

### 4.1. Hypotheses Testing and Structural Model

With the aid of structural equation modeling, the hypothesis was examined. The theoretical basis for the adoption of the structural model was taken into consideration. The statistical method known as structural equation modeling (SEM) can be used to investigate and analyse relationships between observable and patent factors. The overall goodness of fit of the structural equation modeling was assessed using established thresholds from the literature. Figures 2–4 provide a visual representation of the findings.

Figure 2 shows the results of a structural equation modeling study of the data illustrating the relative contributions of the bolstering impact of personality-traits taxonomy (PTT) on operational and environmental performance (OEP). As can be seen in Figure 1 the most significant predictor of personality traits taxonomy (PTT) is openness to experience ($\beta = 0.291$, $p < 0.05$), followed by neuroticism ($\beta = 0.257$, $p < 0.05$), and finally conscientiousness ($\beta = 0.204$, $p < 0.05$). Parameter estimates in Figure 2 suggest that openness to experience is the most significant predictor of operational and environmental performance (OEP). Table 7 indicates the Path coefficients for personality traits, and operational and environmental performance. Considering that every $p$ value was less than 0.05, we then concluded that personality-traits taxonomy (PTT) makes a considerable contribution to operational and environmental performance (OEP). The $p$ values and t values for the variables are shown in Figures 2 and 3, respectively.

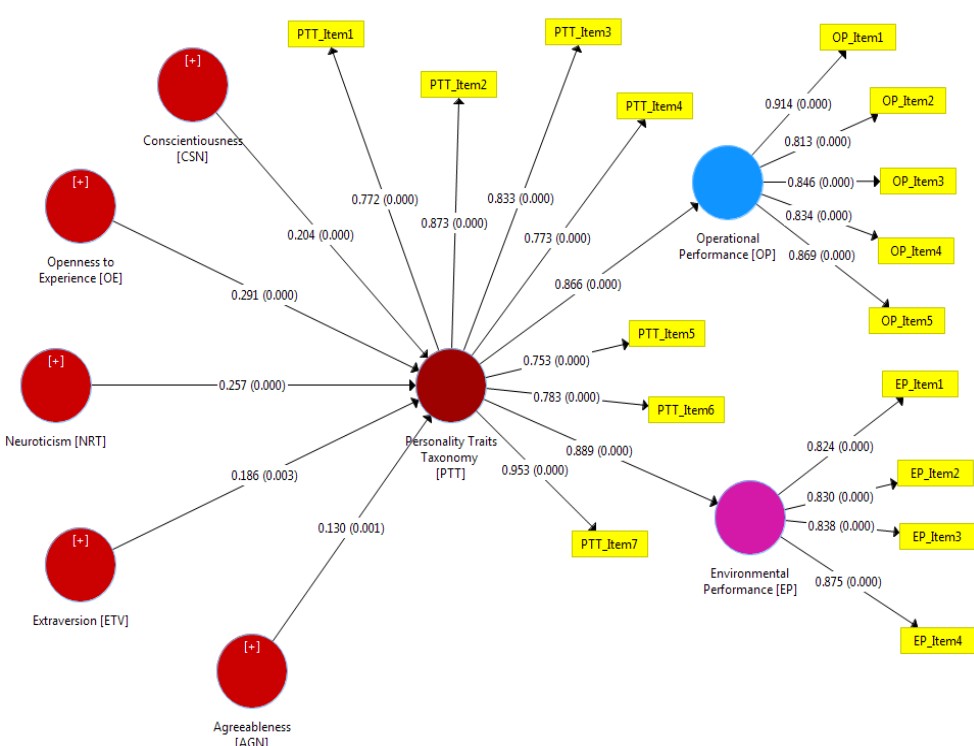

**Figure 4.** SEM path's coefficient and *p*-values.

**Table 7.** Path coefficients for personality traits, operational and environmental performance.

| Variables | Path Co-Efficient | Std. Dev. | *t* Statistics | *p* Values |
|---|---|---|---|---|
| Conscientiousness → PTT | 0.204 | 0.063 | 5.466 | 0.000 |
| Openness to Experience → PTT | 0.291 | 0.078 | 7.237 | 0.000 |
| Neuroticism → PTT | 0.257 | 0.069 | 4.627 | 0.000 |
| Extraversion → PTT | 0.186 | 0.072 | 3.023 | 0.003 |
| Agreeableness → PTT | 0.130 | 0.060 | 3.421 | 0.001 |
| PTT → Operational Performance | 0.866 | 0.073 | 41.501 | 0.000 |
| PTT → Environmental Performance | 0.889 | 0.082 | 53.829 | 0.000 |
| **R Square ($R^2$)** | | | | |
| | R Square ($R^2$) | | R Square ($R^2$) Adjusted | |
| PTT | 0.894 | | 0.880 | |
| PTT → Operational Performance [H1] | 0.749 | | 0.734 | |
| PTT → Environmental Performance [H2] | 0.790 | | 0.781 | |

**Analysis of Hypothesis 1:** *Personality-traits taxonomy do not have influence on the operational performance of SMEs.*

With a (R) value of 0.881, Table 7 gives the model summary of the regression study examining the interplay between personality-trait practices (i.e., conscientiousness, openness to experience, extraversion, neuroticism or emotional resilience, and agreeableness) and the operational performance of manufacturing SMEs. According to the findings, personality-trait practices and manufacturing SMEs' operational performance interact favourably. Additionally, at the 5% level of significance, the interaction's size is statistically significant. Personality-trait practices account for almost 75% variance (i.e., R square) of operational performance among manufacturing small- and medium-sized enterprises in South West Nigeria. According to the findings in Figure 3, the stochastic error term accurately

captures the remaining 25% of fluctuations, which are impacted by other variables not considered in the model. Even after adjusting for the variation explained by other model variables, the standardised beta coefficient shows that personality traits, particularly openness to experience, neuroticism, and conscientiousness, play a substantial role in explaining personality-trait practices across the manufacturing SMEs. Furthermore, the structural model indication of the overall significance (B = 0.866, $R^2$ = 0.749, *p*-value = <0.005) offers compelling evidence that the null hypothesis is rejected. This indicates that the regression model has a strong fit for describing the association between personality-trait practices and the operational performance of manufacturing SMEs, and is highly statistically significant.

**Analysis of Hypothesis 2:** *Personality-traits taxonomy do not have influence on the environmental performance of SMEs.*

With a (R) value of 0.889, Figures 3 and 4 give the model summary of the regression study looking at the interplay between personality-trait practices (i.e., conscientiousness, openness to experience, extraversion, neuroticism or emotional resilience, and agreeableness) and the environmental performance of manufacturing SMEs. According to the findings, personality-trait practices and manufacturing SMEs' environmental performance interact favourably. Additionally, at the 5% level of significance, the interaction's size is statistically significant. Personality-trait practices account for almost 79% variance (i.e., R square) of environmental performance among manufacturing small- and medium-sized enterprises in South West, Nigeria. According to the findings in Figure 2, the stochastic error term accurately captures the remaining 21% of fluctuations, which are impacted by other variables unconsidered in the model. Even after adjusting for the variation explained using other model variables, the standardised beta coefficient shows that personality traits, particularly openness to experience, neuroticism and conscientiousness, play a substantial role in explaining personality-trait practices across the manufacturing SMEs. Furthermore, the structural model indication of the overall significance (B = 0.889, $R^2$ = 0.790, *p*-value ≤ 0.005) offers compelling evidence that the null hypothesis is rejected. This indicates that the regression model has a strong fit in describing the association between the personality-trait practices and environmental performance of manufacturing SMEs and is highly statistically significant. The environmental performance of SMEs in South West Nigeria is significantly impacted by a number of parameters connected to personality-trait techniques. The results show that a 1% improvement in the green supply chain's capacity to reduce overall costs through conscientiousness, openness to experience, extraversion, neuroticism or emotional resilience, and agreeableness will lead to an increase in environmental performance of 79%. This suggests that adopting and promoting returnable packaging practices, participating actively in waste reduction, working with vendors to standardize packaging and deliver directly to users' sites, and using production processes that do not emit harmful substances all improve the performance of manufacturing SMEs.

*4.2. Discussions*

The results of this study confirm the considerable and favourable effects that personality-trait dimensions have on SMEs' operational and environmental performance. The study emphasises the large magnitude of the beneficial relationship between personality-traits taxonomy and manufacturing SMEs' operational and environmental performance. The results of this study are consistent with earlier research by [3,8,12,28]. But it is crucial to keep in mind that personality traits might have an impact on several organisational performance factors. This result also aligns with the submission of [12,28] that personality traits have a significant effect on job performance. This implies that personality traits indirectly enhance operational and environmental performance by having an influence on worker behaviour, decision making, and adaptability.

The findings of [3,12] depict that those who are very concerned about the environment or who have pro-environmental views may be more likely to develop environmentally

friendly practices. These individual-level attitudes and behaviours can potentially influence the overall environmental performance of an organisation. Given the potential impact of personality traits on organisational outcomes, it could be valuable for future research to explore the relationship between personality-traits taxonomy and operational and environmental performance specifically within the context of manufacturing SMEs. This would help to gain a deeper understanding of how individual differences in personality traits may shape the performance and sustainability practices of these types of organisations.

Based on the findings, it was suggested that SMEs in the southwest should be made aware of the important influence and connection between personality-trait activities (including conscientiousness, openness to experience, extraversion, neuroticism or emotional resilience, and agreeableness). Through instructional programmes, the significance of these elements should be highlighted to SMEs operators. Additionally, SMEs must assess personality-trait systems to guarantee operational effectiveness and environmental sustainability.

Theoretical Findings

This study concluded that the personality-traits taxonomy (PTT) of small- and medium-sized enterprises' (SMEs) business owners and managers significantly contributes to operational and environmental performance (OEP). The study specifically supports the assumptions and relevance of the Big Five model and Contingency Theory. This suggests that by altering interventions and strategies based on these traits, SMEs can improve their environmental practices and outcomes. Setting priorities for resource allocation, promoting teamwork and employee involvement, and abiding by laws and industry standards are some examples of how to do this. The Big Five model of personality characteristics provides insightful information on the variables that may affect SME success. Openness, conscientiousness, extraversion, agreeableness, and emotional stability are important personality qualities that SMEs should consider when hiring, developing leaders, and creating a company culture that will increase their chances of success. In the end, applying and comprehending the Big Five model may help SMEs perform better and sustain themselves in a constantly shifting business environment. SME success is a result of a complex interplay between organisational strategies, environmental factors, and the leadership and team-member personalities. While personality traits as defined by the Big Five model shed light on how people approach their roles, contingency theory emphasises the importance of aligning organisational structures and strategies with external contingencies. By combining these viewpoints, SMEs can improve their performance by making strategic decisions that take advantage of their leaders' and teams' strengths while remaining flexible in the face of shifting external circumstances. The sustainability and competitiveness of SMEs in today's dynamic business environment can ultimately be improved using this integrated approach. The main conclusion of the study emphasises the significance of integrating personality traits into management strategies to support sustainable business practices and enhance the operational and environmental performance of SMEs.

## 5. Conclusions and Recommendations

Based on the available information and research, it is evident that the operational and environmental performance in manufacturing SMEs can be influenced by various factors, including but not limited to personality traits. While specific findings regarding the effect of personality-traits taxonomy on operational and environmental performance of manufacturing SMEs are not available, it is important to acknowledge that personality traits can indirectly impact organisational performance through their influence on employee behaviour, attitudes, and decision-making processes.

### 5.1. Recommendations

First, manufacturing SMEs should promote a culture that values employee initiative, innovation, and openness to change in order to enhance operational and environmental

performance. This may be accomplished by promoting information sharing, providing opportunities for training and advancement, and inviting staff input. When selecting employees, and fostering their development, organisations should consider personality traits that are relevant to the desired operational and environmental outcomes.

Second, encouraging employee environmental awareness and responsibility can improve environmental performance. Organisations can provide education and training programmes on sustainability practices, waste reduction, and energy conservation. This can create a sense of ownership and engagement among employees towards environmental sustainability goals.

Third, effective communication and collaboration within the organisation are vital for improving operational and environmental performance. Manufacturing SMEs should foster a collaborative work environment that encourages cross-functional teamwork, knowledge sharing, and information exchange. This can enhance coordination and efficiency in implementing sustainable practices across different operational areas.

Finally, while the specific impact of personality-traits taxonomy on operational and environmental performance of manufacturing SMEs requires further research, implementing the above recommendations can contribute to creating a culture of sustainability, innovation, and operational excellence within these organisations.

### 5.2. Policy and Managerial Implication

Policy and managerial implications play a crucial role in guiding the application of research findings into practical actions. In the context of the study on personality-traits taxonomy and operational and environmental performance in SMEs, these implications provide actionable insights for both policymakers and managers in small- and medium-sized enterprises (SMEs).

i.  Small- and medium-sized enterprises (SMEs) recognise the importance of integrating personality tests into their recruitment processes. This strategic approach facilitates the identification of candidates whose personality traits align with the specific job requirements and the organisational culture. By assessing applicants in terms of their personality traits, SMEs can enhance the likelihood of selecting individuals who not only possess the necessary skills but also exhibit the behavioural characteristics conducive to success within the company.

ii.  SME owners and managers can also benefit from a deeper understanding of their own personality attributes. Self-awareness in leadership is a valuable tool as it allows individuals to identify their strengths and areas for improvement. With this awareness, SME leaders can embark on a more targeted journey of personal and professional development, leveraging their strengths and addressing any potential weaknesses, ultimately becoming more effective in their roles.

iii.  The performance of SMEs can significantly improve by forming diverse teams that encompass complementary personality profiles. Such teams can encompass individuals with diverse traits such as innovative thinking, meticulous attention to detail, effective communication skills, and adaptability. This balance of traits can lead to a more dynamic and productive work environment, fostering creativity, problem-solving, and adaptability.

iv.  SMEs should also recognise the importance of traits like openness to new experiences and emotional stability when navigating change. Managers within these enterprises can play a crucial role in encouraging their staff to embrace change and actively seek opportunities for growth and progress. By fostering a culture that values adaptability and resilience, SMEs can thrive in an ever-evolving business landscape.

v.  To establish and maintain strong customer relationships, SMEs can leverage personality traits such as agreeableness and extraversion. These traits can be invaluable in customer-facing roles, facilitating better communication, empathy, and rapport-building with clients. The result can be increased customer loyalty, positive word-of-mouth recommendations, and a stronger market position.

vi.    Training sessions within SMEs, both for managers and staff members, can focus on developing conflict-resolution techniques, with an emphasis on the agreeableness attribute. Enhanced conflict-resolution skills can lead to improved team dynamics, more amicable working relationships, and a more harmonious work environment. This, in turn, can boost overall productivity and job satisfaction among employees, contributing to the SME's success.

*5.3. Contributions to Knowledge*

The contributions to knowledge in the context of the effect of personality-traits taxonomy on operational and environmental performance of manufacturing SMEs can be summarised as follows:

i.    This research area makes a valuable contribution by shedding light on the potential impact of personality traits on organisational performance, with a specific focus on the manufacturing sector within small- and medium-sized enterprises (SMEs). It underscores the significance of considering individual variations in personality traits when investigating operational and environmental outcomes. This recognition of the role of personality traits offers a more nuanced perspective on how these traits can affect the success and sustainability of SMEs, an area that has been relatively underexplored.

ii.    The research further enriches the field by bridging personality psychology and sustainability studies. By delving into the relationship between personality traits and operational/environmental performance, it uncovers a novel dimension in understanding how individual characteristics can influence sustainable practices within manufacturing SMEs. This integration of seemingly disparate fields offers a more comprehensive view of the factors at play in shaping the sustainability efforts of small- and medium-sized enterprises.

iii.    The practical implications of the study are manifold and extend to manufacturing SMEs. The insights gained from this research can guide various aspects of these organisations, such as recruitment and selection processes, employee development programmes, and overall organisational strategies aimed at enhancing operational and environmental performance. Recognizing the role of personality traits allows SMEs to make informed decisions that contribute to creating a more sustainable and high-performing work environment. By leveraging this knowledge, these enterprises can align their human resource practices and organisational culture with sustainability goals, ultimately driving positive change and competitiveness in the market. The study provides a valuable roadmap for SMEs seeking to incorporate sustainability into their business strategies while considering the individual attributes of their workforce.

*5.4. Limitations and Suggestions for Further Studies*

One limitation is the lack of specific research directly examining the effect of personality-traits taxonomy on the operational and environmental performance of manufacturing SMEs. The absence of targeted studies limits the ability to draw concrete conclusions about the relationship between specific personality traits and these performance outcomes. The generalisability of the findings may be limited due to a lack of studies focusing specifically on manufacturing SMEs. The existing research may have primarily focused on larger organisations or other sectors, which may have different dynamics and resources than SMEs in the manufacturing industry.

Future studies should focus on manufacturing SMEs to better understand the unique dynamics and challenges they face in terms of operational and environmental performance. This would provide more accurate and essential details on the impact of personality-traits taxonomy in this specific case. Through the use of longitudinal research, it would be feasible to assess the long-term consequences of personality traits on operational and environmental performance. We would then have a better understanding of how these qualities emerge and the long-term effects they have on organisational outcomes.

Combining quantitative and qualitative research approaches may enable a greater understanding of the relationship between personality traits and operational and environmental performance. Subtle insights and knowledge about the underlying mechanisms and contextual factors at play may be revealed through qualitative research. Examining possible mediating and moderating elements, such as organisational culture, leadership philosophies, or sector-specific characteristics, might contribute to a fuller understanding of the mechanisms through which personality traits impact operational and environmental performance in manufacturing SMEs.

**Author Contributions:** Conceptualization, C.G.A.; validation and investigation O.P.S.; writing—review and editing, O.O.J. and O.R.A.; project administration, O.O.O.; resources, C.G.A., O.P.S. and O.O.O. All authors have read and agreed to the published version of the manuscript.

**Funding:** Covenant University funded the Article Processing Charges (APC) of this article.

**Institutional Review Board Statement:** The study does not pose any risk to any participant. Therefore, the ethical review and approval were waived.

**Informed Consent Statement:** All the participants gave informed consent for this study.

**Data Availability Statement:** Data are available based on request.

**Acknowledgments:** The authors would like to acknowledge Covenant University Center for Research and Innovation (CUCRID) for sponsoring the publication of this article.

**Conflicts of Interest:** The authors declared no conflicts of interest.

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
