# Peer review of "Personality-Traits Taxonomy and Operational and Environmental Performance: A Cross-Sectional Analysis of Small and Medium Scale Manufacturing Enterprises"

_sustainability, doi:10.3390/su16083497_

Round 1

Reviewer 1 Report

Comments and Suggestions for Authors

Reviewer 2 Report

Comments and Suggestions for Authors

Dear Authors and Editor!

I reviewed the article and made a few comments:

Interesting topic that can be replicated in the study in other contexts.

Summary: good, well-structured, presents the main elements of the research clearly.

Introduction: divided into parts, I prefer a single text, even more so in this case, the text is subdivided only once. The text presents good contextualization. It presents the objectives and hypotheses. The latter I prefer when they come together with the literature in a grounded way. However, they are based on Literature Review.

Literature review: with many subdivided, in a way this organizes the text. They even bring an item Theoretical Justifications, which is not very usual. Recent materials used and within the study context.

Methodology: very detailed item, they made good methodological choices, including the software used for processing. They present some characterization data of the researched population and the actual research data. I missed the explanation of how the questionnaire was prepared and its validation.

Analysis and Discussion: each hypothesis is discussed separately and later discussions are held that are theoretically based. The interesting thing is to highlight the theoretical findings.

Conclusion and Recommendations: It is also subdivided like the other sections of the article. Division makes it easy to find all the items you need in a conclusion. It is pertinently crafted and has good considerations.

Yours sincerely.

Reviewer 3 Report

Comments and Suggestions for Authors

This is a well written paper, with some robust argumentation and interesting findings.

Nevertheless, I have the following major concerns:

1.       Literature review should be more comprehensive. Generally, the manuscript is not sufficiently well-theory informed. In this vein, should include the following paper Meramveliotakis G, Manioudis M. Sustainable Development, COVID-19 and Small Business in Greece: Small Is Not Beautiful. Administrative Sciences. 2021; 11(3):90. https://doi.org/10.3390/admsci11030090.

2.       Discussions and theoretical findings should be expanded.

Comments on the Quality of English Language

 Moderate editing of English language required

Reviewer 4 Report

Comments and Suggestions for Authors

Dear Authors,

The reviewed research paper aims to analyse the influence of the personality trait construct (consisting of conscientiousness, openness to experience, extraversion, neuroticism or emotional resilience, and agreeableness) on the operational and environmental performance of SMEs in Nigeria by using SEM. As stated above, the research theme is interesting in the field of social sciences, although several significant improvements regarding the paper are proposed and needed to be considered:

C1:  Concerning the Introductory part, it should be extended with the originality elements and the structure of the paper, as well merged with the problem statement part. Lines 97-103 should be integrated within the methodological part or hypothesis development, not within the Introduction.

C2: One of the two main shortcomings of the reviewed manuscript concerns the theoretical background of it. This component of the study should be significantly extended with recent studies from the international literature, reflecting a logic and arguments with a critical approach. Thus, a detailed review on studies regarding the individual personality traits, different personality trait taxonomies and SME/company performance (operational, environmental or other types – financial, marketing etc.) should be included. Also, the Authors should discuss what kind of linkages have been already studied within the literature, what kind of results were obtained regarding different personality traits and SME performance.

C3: Regarding the Methodological part the next concerns should be addressed:

- A presentation of the objectives for the quantitative study, the structure (questions/items and measurement scales) of the questionnaire, the primary data collection process.

- Phrases like “This study, which concentrated on all of the registered SMEs manufacturing operators in Southwest,  Nigeria, Nigeria, used a survey research design. A sample size of 420 SMEs were chosen using a stratified random selection approach from the study's population of 1,842 SMEs” should be reformulated, because if the study concentrates on all the Nigerian SMEs, than all of them should be respondents. The Authors should state more clearly that they are in reality the research population.

C4: The results of the quantitative study should be presented more systematically. It is quite hard to follow for the reader pages of the tables followed by figures (e.g. pages 12-13), not formatted in a similar manner, numbered or titled (e.g. Figure 1).

Furthermore, the Authors announced within the Abstract as the objective to test personality traits taxonomy, however no such taxonomy is included within the study, just a simple combination of the five traits within a higher-level construct.

As well, Hypothesis 1 is differently enounced within lines 99-100 and 381. The same applies for hypothesis 2 (lines 102-103 and 410-411). Therefore, a deeper understanding of the literature and a better argument for hypothesis would eliminate this type of errors.

C5: The Conclusions part could be extended, by adding managerial and policy implications.

C6: Some further observations:

- Some typos can be found within the body of the manuscript (e.g. off in line: 57).

- Shortenings should be presented in a longer form first (e.g., PTT line 204).

- The paper overall is not on the template of Sustainability, considering font and format.

- Generally, the manuscript is too divided in short subsections. A division to level 2 is more than enough for research papers.

I hope the above observations will contribute to the improvement of the reviewed manuscript.

Best regards,

  The Reviewer

Comments on the Quality of English Language

Some typos should be eliminated (e.g. off within line 57).

Reviewer 5 Report

Comments and Suggestions for Authors

The present study is relevant for science and practice, as the setting of tasks is practice-oriented and solves the questions of the present, which can be used in the future as part of the development of small and medium-sized business point nevertheless we will outline several recommendations that may contribute to the improvement of the content of the article.

 1. The clear formulation of the Tasks within the study is very helpful. However, we would like to understand -  what are the proposed two hypotheses? What served as this formulation, denote the history of these hypotheses. 

 2. In the section "Discussion)  I would like to hear the opinion of the authors  - how do their resulting studies differ from earlier existing studies on this topic?

3. It should be indicated - what is the scientific practical gain obtained as a result of great empirical research?

4. In conclusion, we would like to understand and hear from the authors what is the volume of data obtained. More specifically, to show what it can affect? Because it turned out to be a very interesting result.

5. To describe how this can have a positive impact if the results of the study are taken into account when developing measures to develop socio-economic processes in the territory.

6. In the introduction, I would like to state whether such studies have been carried out in other countries or in Nigeria?

7. It is recommended to indicate the economic and statistical method used in processing the collected material.

On the whole, it shows a good positive impression and could be published after some refinements. 

Round 2

Reviewer 1 Report

Comments and Suggestions for Authors

1. Briefly describe in Introduction the study design, data collection methods, and data analysis techniques employed.

2. 

In conclusion:

Discuss how their results align with or differ from previous studies in the field.

    Explore potential reasons for discrepancies and highlight the advancements or new insights the research contributes to the existing body of knowledge.

Author Response

Please find attached the corrected version

Reviewer 3 Report

Comments and Suggestions for Authors

The authors addressed all my comments. 

Reviewer 4 Report

Comments and Suggestions for Authors

Dear Authors,

The reviewed manuscript aims to aims to analyse the influence of the personality trait construct (consisting of conscientiousness, openness to experience, extraversion, neuroticism or emotional resilience, and agreeableness) on the operational and environmental performance of SMEs in Nigeria by using SEM. As stated above, the paper’s theme is interesting, however even after the second round of review, some further improvements are proposed and needed to be considered, because the majority of the previous review observations and suggestions were either not considered or have been limitedly addressed:

C1: The paper overall is not on the template of Sustainability, considering font and format.

C2: Concerning the Introductory part, lines 99-106 (hypotheses H1 and H2) should be rather included within the Methodological part.

C3:  The theoretical background of the considered manuscript should be further extended. This component of the study should be extended with recent studies from the international literature, reflecting a logical presentation and arguments with a critical approach. Complementary, it should include a presentation of the research model (2.3). Also, the Authors should discuss what kind of linkages have been already studied within the literature, what kind of results were obtained regarding different personality traits and SME performance.

C4: Concerning the methodological part, a presentation of the objectives for the quantitative study, the structure (questions/items and measurement scales) of the questionnaire, the primary data collection process must be included.

C5: The results of the quantitative study should be reorganized and detailed in a logical manner: tables and figures explained immediately after including them in the paper.

C6: Elements from subsections 5.2 and 5.3 should be slightly extended, by including phrases and not just some points.

Hope the above observation will contribute to the improvement of the reviewed manuscript.

Best regards,

  The Reviewer

Author Response

(The authors gave the same response as above.)
